# Scaling Risk Assessment in Nanofiltration of Mine Waters

**DOI:** 10.3390/membranes10100288

**Published:** 2020-10-15

**Authors:** Krzysztof Mitko, Ewa Laskowska, Marian Turek, Piotr Dydo, Krzysztof Piotrowski

**Affiliations:** 1Department of Inorganic, Analytical Chemistry and Electrochemistry, Faculty of Chemistry, Silesian University of Technology, ul. B. Krzywoustego 6, 44-100 Gliwice, Poland; marian.turek@polsl.pl (M.T.); piotr.dydo@polsl.pl (P.D.); 2PolymemTech Sp. z o.o., Al. Niepodległości 118/90, 05-577 Warsaw, Poland; laskowska.ewi@gmail.com; 3Department of Chemical Engineering and Process Design, Faculty of Chemistry, Silesian University of Technology, ul. M. Strzody 7, 44-100 Gliwice, Poland; krzysztof.piotrowski@polsl.pl

**Keywords:** membrane module modeling, calcium sulfate precipitation risk, ionic rejection coefficients

## Abstract

Nanofiltration can be applied for the treatment of mine waters. One of the main problems is the risk of crystallization of sparingly soluble salts on the membrane surface (scaling). In this work, a series of batch-mode nanofiltration experiments of the mine waters was performed in a dead-end Sterlitech^®^ HP 4750X Stirred Cell. Based on the laboratory results, the concentration profiles of individual ions along the membrane length in a single-pass industrial-scale nanofiltration (NF) unit was calculated, assuming the tanks-in-series flow model inside the membrane module. These calculations also propose a method for estimating the maximum achievable recovery before the occurrence of the calcium sulfate dihydrate scaling in a single-pass NF 40″ length spiral wound module, simultaneously allowing metastable supersaturation of calcium sulfate dihydrate. The performance of three membrane types (NF270, NFX, NFDL) has been evaluated for the nanofiltration of mine water.

## 1. Introduction

Nanofiltration (NF) and reverse osmosis (RO) are widely used for water and wastewater treatment. Nanofiltration, due to its high rejection of all ions except the monovalent ones, is widely used to eliminate the groundwater hardness [1] or to recover important substances, such as proteins and sugars. NF may also be considered as an alternative to RO for desalination of brackish waters, where SO_4_^2−^ is the prevailing anion [2]. Nanofiltration is also used as a pretreatment [3] before other methods, such as reverse osmosis, as well as for the treatment of various mine waters, including acidic mine waters [4,5], discharge from oil sand mining [6], saline waters from the mining industry [7].

With the increase in the permeate recovery, concentration polarization rises, increasing the probability of membrane fouling (especially in the form of scaling) because of increasing concentration of sparingly soluble substances in the close vicinity of the membrane surface (e.g., CaSO_4_·2H_2_O, CaSO_4_·1/2 H_2_O, CaSO_4_, BaSO_4_, SrSO_4_, CaCO_3_, SiO_2_, etc.) [8,9,10,11]. Thus, scaling leads to significant degradation of membrane performance, shortening of membrane lifetime, decrease in permeate quality, increase in the energy consumption as higher pressure difference is necessary, increased cleaning frequency, higher consumption of antiscalants, and in the worst cases, irreversible membrane degradation. Antiscalant treatment and/or pH adjustment are usually used as the feed water pretreatment methods to decrease the mineral scale formation [12,13]. Various cleaning procedures and surface modifications are also applied in membrane scaling control [14]. An accurate prediction and monitoring of scaling probability and its progress are important. In practice, all these methods are usually applied together to decrease or eliminate the membrane scaling consequences [14]. From the economic and environmental points of view, the earliest possible detection of the scaling onset is crucial. To estimate the calcium sulfate dihydrate solubility in various electrolyte solutions, a reference state for different supersaturated systems, different methods based on specific correlations for the activity coefficients that include the Debye–Hückel, Guggenheim–Davies expressions, Bromley, Meissner or Pitzer models are proposed [15].

One possible approach to prevent scaling formation is to design the membrane process taking into account the hydrodynamic apparatus conditions and scaling kinetics. This approach has been used in the electrodialysis of waters having high scaling potential [16,17].

In the presented work, a methodology of estimating the maximum allowable permeate recovery in the nanofiltration module is presented, with the aim of meeting the needs of increasing the permeate recovery and the process safety, in particular for the application of nanofiltration membranes for the treatment of mine waters. Based on the own laboratory test results and model calculations, the ion concentration profiles along a single-pass industrial-scale NF 40″ length spiral wound membrane element were estimated, assuming the tanks-in-series reactor model describing the module performance. These calculations allowed estimating the maximum allowable recovery still preventing the calcium sulfate dihydrate coupled nucleation and scaling occurrence on the membrane surface established in this single-pass NF module construction. The software provided by the manufacturers typically predicts final parameters of the retentate/permeate; however, modeling the nanofiltration in the manner presented in the manuscript can give additional information, such as concentration profile along the membrane, the place inside the module where the scaling risk increase.

## 2. Materials and Methods

### 2.1. Chemistry of Raw Water

Investigation of scaling risk assessment in nanofiltration membranes (Table 1) was carried out using two different mine waters of different composition, denoted as “A” and “B”. Mine water “A” (representing the brackish water) has total dissolved salts (TDS) content of 1.8 g/L and has higher sulfate ions concentration than the chloride ones. Real samples collected at the premises of mines were used. The mine water “B” (representing the brine) has TDS of 55 g/L and significantly higher chloride ions concentration compared to sulfates one. The original pH of mine water “A” was adjusted to 5.7 before the experiment to prevent the CaCO_3_ scaling. The carbonate ions are relatively easy to remove in the plant pretreatment (i.e., by decarbonization using acid dosing or weak ion exchangers). As such, the focus was put on a more difficult scale-forming compound, calcium sulfate. Ionic composition of each mine water sample was determined using ion chromatography (ICS-5000 Thermo Dionex, Waltham, MA, USA). Concentrations of the main ions are presented in Table 1.

### 2.2. NF Membranes

Three commercially available nanofiltration membranes—NF270 (Filmtec), NFX (Synder), and NFDL-5 (Suez) were tested in this study. The properties of these NF membranes are summarized in Table 2. All of the used membranes are thin-film composite polyamide and are negatively charged at the experimental conditions.

### 2.3. Experimental Procedure

Because the mine water samples contained a large amount of total suspended solids, preliminary purification was necessary. Microfiltration (microfilter with pore size 0.45 μm) was used as a pretreatment for “A” and “B” mine waters. Nanofiltration experiments were carried out in a laboratory-scale dead-end Sterlitech^®^ HP 4750 Stirred Cell stainless steel membrane module equipped with a cooling jacket to keep the stable module temperature set at 21 °C. The commercial flat sheet nanofiltration membranes were cut into circular-shaped pieces, with an effective membrane area of 14.6 cm^2^, and then used in all NF procedures (for every single experiment, some new, “fresh” piece of the membrane was applied). Compressed argon was used as a pressure difference source, and the applied pressure was 40 bar. 

Each experiment consisted of the following steps:Place the freshly cut membrane in the membrane module.Pour 300 mL of deionized water (Millipore Elix 10 system, conductivity 0.066 µS) into the feed/retentate chamber of the membrane module.Start the filtration; note the time required to collect every 30 mL of permeate. If the pure water flux at constant pressure doesn’t change between each 30 mL of permeate collected, the membrane is assumed as conditioned. If not, go back to step 2.Pour out the retentate, fill the feed/retentate chamber with the tested solution.Start the filtration and collect 30 mL of permeate. Pour out the retentate and permeate out; fill the feed/retentate chamber with 300 mL of tested solution.Start the filtration and collect 90 mL of permeate. Stop the filtration, recycle the collected permeate back to the feed/retentate chamber, and collect the feed sample for analysis.Start the filtration. Collect each 30 mL of permeate into a separate sample container.

Ionic composition of all solutions (permeate, feed, retentate after finished experiments) was determined using ion chromatography (ICS-5000 Thermo Dionex, Waltham, MA, USA). Experimental rejection of i-th ion was calculated using the Equation (1):R_i_ = (1 − C_p,i_/C_f,i_)·100%,(1)
where C_p,i_ and C_f,i_ are the concentrations of the i-th ion in permeate and feed, respectively.

### 2.4. Modeling

The idea is to use the rejection coefficients observed in the dead-end filtration in the modeling of the cross-flow filtration. The necessary condition is to create hydrodynamic conditions which assure the concentration polarization is negligible and does not affect the conditions during the batch studies. For instance, in other studies that use the dead-end filtration, it was observed that the stirred cell, the concentration polarization became negligible at rpm > 100 and Re > 12,000 [19]. To assure minimized concentration polarization, we have used a very high-speed mixer (1200 rpms)—which, together with the dimensions of the system, resulted in turbulent flow and good mixing near the membrane surface (57,522 < Re < 60,802, 836 < Sc < 883, 1533 < Sh < 1569—the calculations are presented in Appendix C). It should be stressed, however, that the mixing speed at which the concentration polarization becomes minimized strongly depends on the design and size of the tank and the mixer. The effect of stirrer design on the mixing is particularly important at very turbulent flows; for instance, above Re > 10,000 increasing the Reynolds number have no effect on Power number, but the impeller geometry can still significantly affect the quality of mixing.

The purpose of the proposed method is to estimate the borderline conditions for the high-recovery nanofiltration of the waters having high scaling risk. Although the batch-mode nanofiltration works in a discontinuous unsteady state and doesn’t fully translate into a steady-state single pass nanofiltration, the presented approach could be used to set the boundaries for a pilot-scale verification of the high recovery nanofiltration, such as which membrane to use, what recovery limit should not be crossed, how to position scaling detectors (e.g., ultrasonic ones) along the module for the detection of scaling onset, etc.

To simulate the performance of a spiral-wound NF module, it was assumed that a single feed/retentate channel has a cuboid shape, i.e., any folding of a feed/retentate channel is neglected. The flow channel can then be conventionally regarded as a set of elementary units for each of the shapes depicted in Figure 1—a cuboid of a height h, length Δl, and of width s. The liquid flows between the NF membranes, with the permeate flowing vertically to the direction of the feed flow.

Mass balance of the elementary unit is given as (total—Equation (2), for individual *i*-th ion—Equation (3)):ρ_r_(n)·V_r_(n) = 2 ρ_p_(n)·V_p_(n) + ρ_r_(n + 1)·V_r_(n + 1),(2)
C_r,i_(n)·V_r_(n) = 2 C_p,i_(n)·V_p_(n) + C_r,i_(n + 1)·V_r_(n + 1),(3)
where ρ and C represent the solution density (r—retentate, p—permeate), and i-th ion molar concentration, respectively. Volumetric flow rate of permeate V_p_ can be expressed in terms of the permeate flux J_v_ providing Equation (4):V_p_(n) = J_v_(n)·h·Δl(4)

Assuming constant density (ρ_r_ = ρ_p_ = ρ), symmetric permeate outflow in respect to both parallel NF membrane planes and taking into account the assumed dimensions of the elementary unit (Figure 1), the Equations (2)–(4) can be rearranged to Equations (5) and (6):u_r_(n) = 2 J_v_(n)·Δl/s + u_r_(n + 1)(5)
C_r,i_(n)·u_r_(n) = 2 C_p,i_(n)·J_v_(n)·Δl/s + C_r,i_(n + 1)·u_r,i_(n + 1)(6)
where u is the retentate linear flow rate, defined as Equation (7):u_r_(n) = V_r_(n)/(hs),(7)

Permeate recovery, Y, and rejection coefficient of i-th ion, R_i_, are defined as (where 0—inlet, n—n-th elementary unit)—Equation (8):Y(n) = [u_r_(0) − u_r_(n)]/u_r_(0) = u_p_(n)/u_r_(0),(8)
R_i_(n) = [C_r,i_(n) − C_p,i_(n)]/C_r,i_(n) = f (Y, C_r_,_i_),(9)

The rejection coefficients of the ions depend on the permeate recovery, membrane type, and in this case study, feed water composition (mine water “A” and “B”). Substitution of Equation (9) into Equation (6) results in a set of equations making the calculation of linear flow rate and the individual ions concentration profiles along the membrane length knowing appropriate starting values at the module inlet (n = 0) possible—Equations (10) and (11):V_r_(n + 1) = V_r_(n) − 2 J_v_(n)·Δl·h,(10)
C_r,i_(n + 1) = {V_r_(n)·C_r,i_(n) − 2·J_v_(n)·Δl·h C_r,i_(n)· [1 − R_i_(n)]}/[V_r_(n + 1)],(11)

Each n-th elementary unit was treated as a separate entity, with all the inflow originating from the elementary unit (n − 1) and the outflow going directly to the elementary unit (n + 1)—it was assumed that no back-mixing or longitudal dispersion exists to simulate the plug flow conditions.

To assess the membrane scaling risk, the saturation level of a given sparingly soluble salt has to be considered at the membrane surface facing the retentate channel instead of the saturation level at the retentate bulk. The following concentration polarization profile was assumed—Equation (12):C^m^_r,I_ = C_r,i_·exp[J(n)/k],(12)
where k denotes the mass transfer coefficient at the wall (the membrane), calculated using Equation (13) [20]:Sh(n) = k(n)·d_h_/D_i_ = 1.85 [Re(n)·Sc(n)·d_h_/s]^1/3^,(13)
where Sh, Re, Sc denote the Sherwood, Reynolds, and Schmidt dimensionless numbers, d_h_ is the hydraulic diameter of the channel, s is the channel’s length, and D_i_ is the diffusion coefficient of the *i*-th ion—calculated according to procedure described in Reference [21]. Knowing the predicted ions related directly to at the membrane surface, gypsum saturation level σ may be calculated with Equation (14) [22]:(14)$$\sigma =a_{Ca^{2+}}\cdot a_{{SO_{4}}^{2-}}\cdot (a_{H_{2}O}{)}^{2}/K_{\mathrm{sp}},$$
where ai is the activity of *i*-th ion, based on the Bromley equation for a high ionic strength solution [23]; and K_sp_ is the solubility product of the calcium sulfate dihydrate, calculated with Equation (15) for a given process temperature T [K] [24]:ln(K_sp_) = 390.9619 − 152.624 log(T) − 12545.62/T + 0.0818493 T,(15)

Nucleation induction time t_ind_ of calcium sulfate dihydrate was calculated with the semi-empirical Equation (16) [22,24]:t_ind_ = K·σ^−r^,(16)
where the constants K = 1.3 × 10^5^ s, and r = 5.6 [22].

Geometric parameters of the experimental test module under study were as follows: total module length, L: 0.916 m; length of elementary unit assumed for the calculations, Δl: 0.001 m; intermembrane distance, s: 7.87 × 10^−4^ m (31 mil spacer).

Scaling indices were estimated using the Phreeqc aqueous phase thermodynamic modeling package from the U.S. Geological Survey [25]. The Phreeqc software uses the extended Debye–Huckel and the Davies equation to model the activity coefficients in the liquid phase. Scaling potential of both mine waters, A and B, was confirmed using the ROSA package from DOW Filmtec [26]. The ROSA software simulates the membrane treatment operations based on empirically determined separation factors for different ions and under the given operating conditions.

## 3. Results and Discussion

### 3.1. Nanofiltration

Laboratory tests were carried out in a Sterlitech^®^ HP 4750 Stirred Cell membrane module. The individual effect of permeate recovery Y on SO_4_^2−^, Cl^−^, Ca^2+^, Mg^2+^, and Na^+^ rejection coefficients R_i_ for both analyzed waters “A” and “B”, as well as for three types of NF membranes (NFX, NF270, and NFDL), are presented in Appendix A. Based on the laboratory results, the rejection coefficients were calculated with Equation (1), and a set of empirical equations correlating the resulting rejection coefficients R_i_ of common ions with permeate recovery Y [%], and ion concentration in the elementary cell (n − 1), C_i_ [mg/dm^3^], were established—see Appendix B.

The rejection coefficients for brackish water “A” are considerably higher (for all ions considered) than for brine “B”. This may be caused by the different composition of each feed water. The water “A” contains less chloride, sodium, calcium, and magnesium ions, which causes a smaller diffusion driving force across the membrane, resulting in lower flux of these ions across the membrane; simultaneously, the water flux across the membrane is higher in the case of less saline water “A”, causing overall lower rejection coefficients of calcium and magnesium. This effect does not happen in the case of the sulfate, as both waters contain similar amounts of this ion. The rejection of sulfate was significantly higher than the rejection of calcium and magnesium, which was caused by the negative surface charge of the membrane at the experimental conditions (pH ≥ 5.7).

The highest chloride rejection was observed when the NFDL membrane was used. Moreover, in most cases, NDFL membrane type demonstrates the highest rejection of both univalent cations and bivalent cations, while the NFX membrane generally shows the lowest rejection among all three types studied. The results are in line with the ionic rejection coefficients previously reported in the literature; for instance, Hilal et al. [27] reported achieving low rejection coefficients of monovalent ions and high rejection coefficients of multivalent ions when applying nanofiltration with polyamide membranes, including the NF270, to process concentrated solutions. Kelewou et al. [28] achieved similar results using polyamide-based membranes, including the NF270 membrane used in the presented experiments. They have concluded that the chloride ion is mostly transported through the nanofiltration membrane by diffusion, while the sulfate ion was mostly removed by the convection.

### 3.2. Scaling

The correlations obtained from laboratory data were then used to calculate the concentration profiles of the considered 5 ionic species along the membrane module length arranged in a single-pass industrial-scale NF unit, assuming the tanks-in-series reactor model of a flow inside the membrane module. These calculations allowed to estimate the maximum allowable recovery that would effectively prevent the membrane surface scaling phenomena in a single-pass NF module. Figure 2, Figure 3, Figure 4, Figure 5, Figure 6 and Figure 7 show the calcium sulfate dihydrate saturation profiles along the simulated NF membrane length for the assumed: 65%, 70%, 75%, 80%, 85%, and 90% of the permeate recovery, Y. Saturation of calcium sulfate dihydrate increases along the membrane module length with the permeate recovery, Y in all considered cases. However, the nonlinearity effect is different depending on the Y parameter assumed, processed solution and membrane type. Mine water “A” is supersaturated as early as at 65% of the permeate recovery, while mine water “B” becomes supersaturated later—starting from 75% recovery. Only for NFDL membrane type and “B” mine water, supersaturation of calcium sulfate dihydrate rises sharply exceeding over 600% as early as at 90% recovery. In other cases, at 80% (and lover) recovery, it is typically below 300%. In the practical applications, it is usually assumed that while the solution becomes supersaturated at calcium sulfate saturation of 100%, but it is safe to operate nanofiltration modules up to saturation of ca. 160–200% at best, due to the wide metastable zone of calcium sulfate. There are known examples of operating nanofiltration modules at 300%–400% of calcium sulfate saturation [29], but 600% is way beyond any safety limits of water treatment operations, as it indicates immediate and severe scaling on the membrane surface. However, to assess the scaling severity, one should also take into account the nucleation kinetics and residence time of the supersaturated solution.

Table 3 and Table 4. Show the induction time values for calcium sulfate dihydrate in the retentate corresponding to the nanofiltration conditions under study and theoretical time needed for the solution to flow last 30 cm of the module at: 65%, 70%, 75%, 80%, 85%, or 90% recovery, appropriately. One should keep in mind, however, that these results are valid for nanofiltration working at 40 bar of hydraulic pressure. As the pressure can influence the rejection coefficients, the batch mode experiments should be repeated if this method is to be applied for different hydraulic pressure.

To assess the scaling risk on the membrane surface following methodology is proposed, based on the earlier research [16,17]:Calculate the permeate recovery Y of the module by assuming the feed linear flow velocity (refer to Section 2.3 Modeling).Assume a point along the module membrane length.Calculate the bulk retentate ionic concentrations and retentate concentrations at the membrane surface for a chosen point using the previously discussed model.Calculate the theoretical time needed for the solution to flow from the chosen point to the retentate outlet. The gradual change in the volumetric flow along the membrane module length, due to the flow across the membrane is taken into account when calculating the time needed to leave the module has been taken into account by calculating the mean residence time in each of the elementary units separately and adding them.If the theoretical time needed for the solution is not at least six times higher than the induction time, the scaling risk is unacceptablely high.

Using the above assumptions, a maximum allowable recovery was calculated for each feed water and membrane type, defined as the maximum permeate recovery for which there is no unacceptable high risk of scaling at any point along the membrane module length—the results are presented in Table 4.

The results suggest that performing the nanofiltration at high permeate recovery (even above 90%) should be possible without the scaling on the membrane surface. One should remember, however, that in reality the supersaturated solution doesn’t leave the module precisely at the last point along the membrane and can stay for some time in the piping. In practical operation, a lower permeate recovery value (~ 85%) would be more recommended, as well as placing the precipitator for supersaturated retentate immediately after the pressurized device, to avoid unnecessary holdup in the piping.

The scaling risk is the highest when applying the NFDL membrane and the lowest when applying the NFX membrane. This can be explained as a result of different chemistry of the obtained retentate: The NFDL membrane has shown the highest rejection coefficients of bivalent ions during the bench-scale tests, which means the retentate obtained using this membrane is the most supersaturated with calcium sulfate. On the other hand, NFDL is more hydrophobic than NF270, so it may show less tendency for scale layer growth in the same saturation conditions.

## 4. Conclusions

Based on the permeate flux and sodium, magnesium, calcium, chloride, and sulfate ions concentration measurements in the dead-end experiments, the scaling risk of calcium sulfate dihydrate in the NF 40″ length spiral wound membrane module was estimated. The dead-end experiments showed that the nanofiltration process may be safely operated even at 80% recovery of permeate. A method of predicting the operational limits of nanofiltration modules working in high scaling risk situations, e.g., when the feed water is rich in calcium and sulfate, was proposed. Comparing the theoretical time needed for solution to flow through the module and calculated nucleation induction time of calcium sulfate dihydrate for a given final retentate concentration, it is possible to predict maximal safe recovery level Y for each specific process conditions; the established model, however, is valid only at given hydraulic pressure (40 bar) and would need an additional set of experiments to include pressure as a variable. Experimental tests clearly demonstrated that scaling-free operation of the 40″ length spiral wound NF module is possible at 75% to permeate recovery in the case of highly concentrated mine water and with 80% permeate recovery considering brackish water. A possibility of working at 75% to permeate recovery level should improve the performance of the integrated salt production systems using NF as pretreatment step, since they are limited in terms of overall recovery by the pretreatment (NF) recovery. The establishment of the reliable and mathematical model to simulate the nanofiltration in large-scale systems creates an opportunity for the investigation of NF applicability in several technologically important processes.

## Figures and Tables

**Figure 1 membranes-10-00288-f001:**
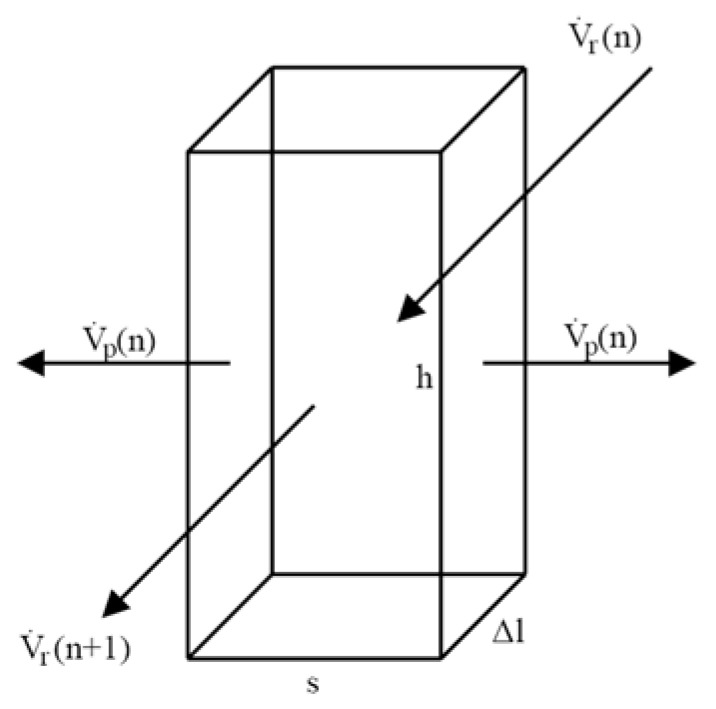
Scheme of an elementary NF module unit.

**Figure 2 membranes-10-00288-f002:**
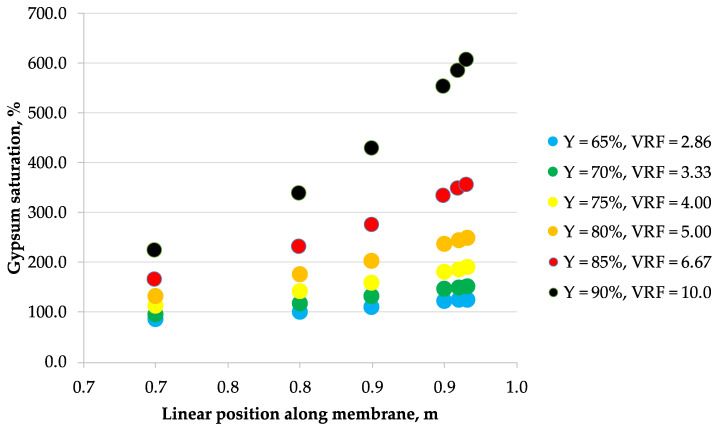
Saturation vs. position along the membrane for mine water A (0.384 g/L as Cl^−^, 1.02 g/L as SO_4_^2−^, 0.107 g/L as Na^+^, 0.142 g/L as Mg^2+^, 0.312 g/L as Ca^2+^) and NF270 nanofiltration membrane.

**Figure 3 membranes-10-00288-f003:**
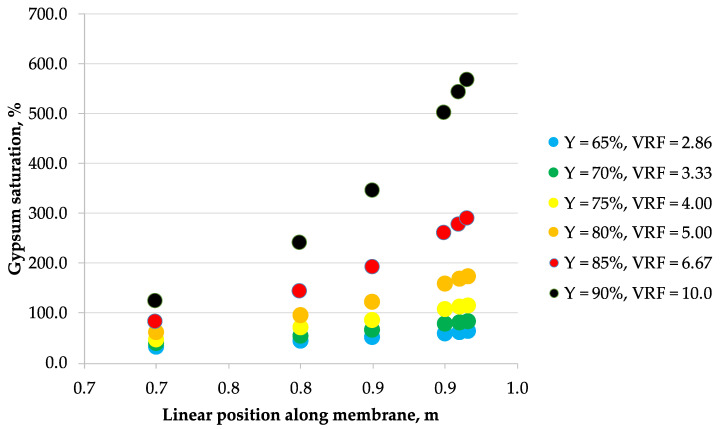
Saturation vs. position along the membrane for mine water B (33.2 g/L as Cl^−^, 0.937 g/L as SO_4_^2−^, 19.5 g/L as Na^+^, 0.990 g/L as Mg^2+^, 0.771 g/L as Ca^2+^) and NF270 nanofiltration membrane.

**Figure 4 membranes-10-00288-f004:**
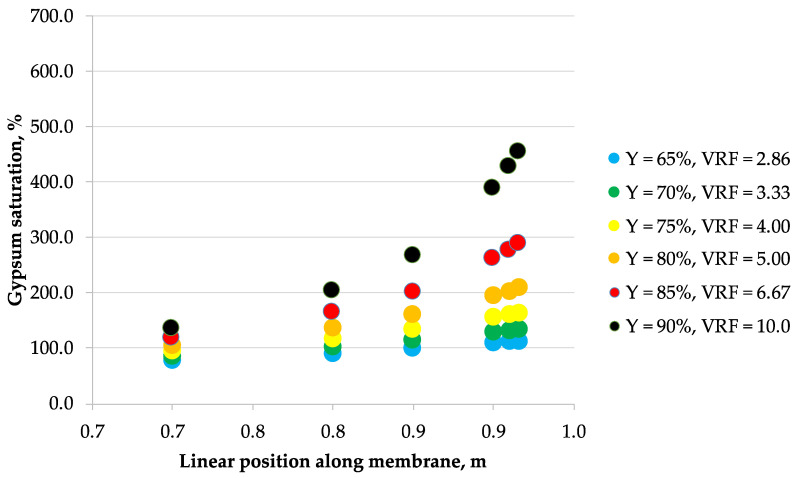
Saturation vs. position along the membrane for mine water A (0.384 g/L as Cl^−^, 1.02 g/L as SO_4_^2−^, 0.107 g/L as Na^+^, 0.142 g/L as Mg^2+^, 0.312 g/L as Ca^2+^) and NFX nanofiltration membrane.

**Figure 5 membranes-10-00288-f005:**
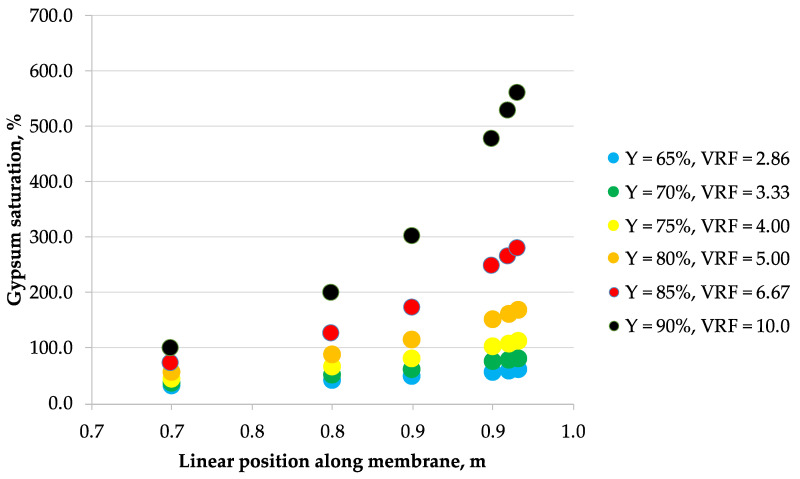
Saturation vs. position along the membrane for mine water B (33.2 g/L as Cl^−^, 0.937 g/L as SO_4_^2−^, 19.5 g/L as Na^+^, 0.990 g/L as Mg^2+^, 0.771 g/L as Ca^2+^) and NFX nanofiltration membrane.

**Figure 6 membranes-10-00288-f006:**
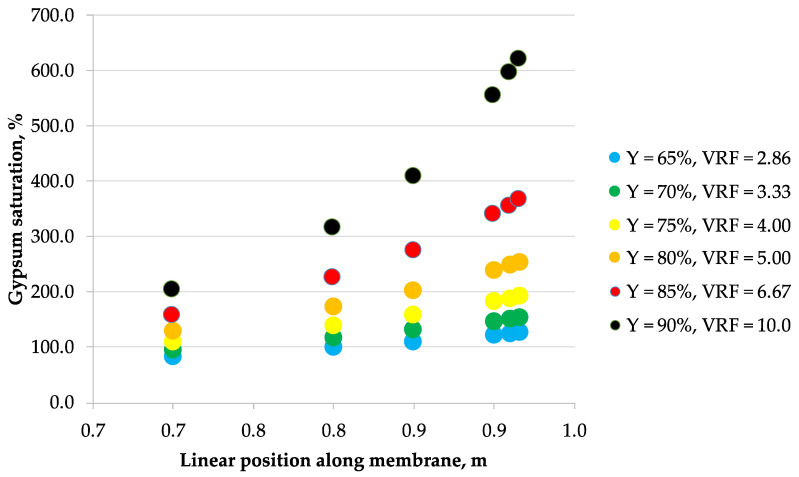
Saturation vs. position along the membrane for mine water A (0.384 g/L as Cl^−^, 1.02 g/L as SO_4_^2−^, 0.107 g/L as Na^+^, 0.142 g/L as Mg^2+^, 0.312 g/L as Ca^2+^) and NFDL nanofiltration membrane.

**Figure 7 membranes-10-00288-f007:**
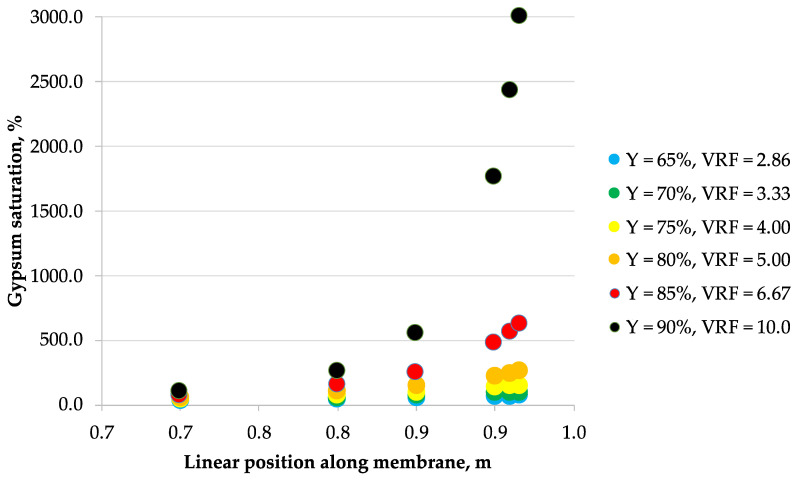
Saturation vs. position along the membrane for mine water B (33.2 g/L as Cl^−^, 0.937 g/L as SO_4_^2−^, 19.5 g/L as Na^+^, 0.990 g/L as Mg^2+^, 0.771 g/L as Ca^2+^) and NFDL nanofiltration membrane.

**Table 1 membranes-10-00288-t001:** Properties of the nanofiltration (NF) membranes were applied.

Mine Water	Langelier Saturation Index (LSI)	Gypsum Saturation	Concentration, g/L				
			Cl^−^	SO_4_^2−^	Na^+^	Mg^2+^	Ca^2+^
A	−2.4	34%	0.384	1.02	0.107	0.142	0.312
B	−2.0	11%	33.2	0.937	19.5	0.990	0.771

**Table 2 membranes-10-00288-t002:** Properties of the NF membranes were applied.

	NFX ^1^	NFDL ^1^	NF270 ^1^
Supplier	Synder Filtration	Suez	Dow FilmTec
Maximum Operating Temperature, °C	50	50	45
pH range	3–10.5	3–9	2–11
Minimum MgSO_4_ rejection, %	99	96	99.2
Membrane material	Polyamide thin-film composite	Polyamide thin-film composite	Polyamide thin-film composite
Isoelectric point	3.2	4	3.0
Molecular weight cut-off, Da	150–300	150–300	200–400
Average pore width, nm	n/a	9.6 [18]	7.9 [18]
Contact angle, °	n/a	37.9 [18]	15.9 [18]

^1^ Test conditions according to membrane supplier information: 2000 ppm MgSO_4_ inlet solution at 110 psi (760 kPa) operating pressure, isothermal process conditions at 77 °F (25 °C), tests at 15% permeate recovery after 24 h of filtration.

**Table 3 membranes-10-00288-t003:** Induction time of calcium sulfate dihydrate during the NF process at the retentate outlet (0.916 m of module length)—effect of permeate recovery Y, mine water (“A”: 0.384 g/L as Cl^−^, 1.02 g/L as SO_4_^2−^, 0.107 g/L as Na^+^, 0.142 g/L as Mg^2+^, 0.312 g/L as Ca^2+^; “B”: 33.2 g/L as Cl^−^, 0.937 g/L as SO_4_^2−^, 19.5 g/L as Na^+^, 0.990 g/L as Mg^2+^, 0.771 g/L as Ca^2+^) and nanofiltration membrane type.

Y, %	t_ind_, s					
	NF270 “A”	NFX “A”	NFDL “A”	NF270 “B”	NFX “B”	NFDL “B”
65	28,560	48,900	27,180	1,156,440	1,403,220	657,900
70	11,100	20,760	10,440	283,320	348,900	126,780
75	3648	7560	3384	53,340	65,880	15,540
80	930	2202	846	7080	8520	894
85	153.6	447	136.8	560.4	648	12.24
90	10.8	46.62	9.18	19.2	20.7	4.60 × 10^−3^

**Table 4 membranes-10-00288-t004:** The maximum allowable recovery for each membrane type and feed water (“A”: 0.384 g/L as Cl^−^, 1.02 g/L as SO_4_^2−^, 0.107 g/L as Na^+^, 0.142 g/L as Mg^2+^, 0.312 g/L as Ca^2+^; “B”: 33.2 g/L as Cl^−^, 0.937 g/L as SO_4_^2−^, 19.5 g/L as Na^+^, 0.990 g/L as Mg^2+^, 0.771 g/L as Ca^2+^).

	Membrane Type					
	NF270 “A”	NFX “A”	NFDL “A”	NF270 “B”	NFX “B”	NFDL “B”
Maximum allowable recovery [%]	90.1	91.6	89.8	89.3	89.5	84.6

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
