# Peer review of "Scaling Risk Assessment in Nanofiltration of Mine Waters"

_membranes, 2020, doi:10.3390/membranes10100288_

Round 1

Reviewer 1 Report

The measurement procedure with the dead-end test cell is not clearly described. From the description of the methods it is not clear to me how these data were determined. The revision of this description of the practical measurements with the test cell seems to me to be absolutely necessary.

Not to include carbonates here cannot be justified by setting the pH to a value of 5.7. Carbonates and sulphates are the essential ions for possible precipitation or scaling and carbonate ions should also be considered here. Chlorides, for example, are highly soluble and seems to me meaningless in such a study, as they only make a small contribution to flow resistance and osmotic pressure. Why are chlorides also described here.

The notion of a recovery Y, to which the retention in the test cell is referred to here, seems unusual for such tests. A factor that reflects the volumetric concentration of the feed solution introduced into the test cell should be more appropriate here. Furthermore, the specific permeate flows to be observed in these measurements of the retention should be indicated.

With a membrane area of 14.6 cm2 and a pressure of 40 bar I estimate these permeate flows in the range of 1 mL/min. The starting volume in the test cell is also not given. If 100 mL were to be filled, such a concentration experiment, in which 90 % of the feed solution is obtained as permeate, would take about 2 h, whereby the concentrations in the retentate change over time. In a technical plant for water treatment, the operation is normally continuous and the concentrations and flows are locally constant. The experimental data are taken here from discontinuous unsteady experiments.  

It has to be explained why these data are suitable for statements on continuous separation processes and here for the assessment of the maximum possible recovery in operating plants.

Line 85:  heady load ?

Line 22: twice NF270 !

Line 67: NFDL-5 is originally a Desal-Membrane, than GE-Osmonics and now a Suez membrane product

Line 158: Why it's a Stirred Cell membrane module. Such a stirred cell contains a flat membrane and a membrane module is a technical concept with a specific configuration for the application.

Line 203 to 206: Here the laboratory data are transferred to a continuous separation process with a 40” spiral module (as stated above). How is this module operated, what is the procedure here and why is this permissible or acceptable and applicable?

Line 279: I don't think that a redesign of the module is the task here.

Line 282: …it is possible to determine possibly…seems to me a speculation and not a conclusion!

Line 287: Explain the overall recovery newly mentioned here. Why this extrapolation is possible. It seems more like speculation here.

Line 291: “…possible precise…”  Is it or not and what means then precise here.

Author Response

Thank you for the thorough review. We tried to fix the issues we mentioned and we hope the manuscript is better now. Please find below the detailed response to your questions.

“The measurement procedure with the dead-end test cell is not clearly described. From the description of the methods it is not clear to me how these data were determined. The revision of this description of the practical measurements with the test cell seems to me to be absolutely necessary.”

We revised the description of the experimental procedure in hope that it’s more clear now.

“Not to include carbonates here cannot be justified by setting the pH to a value of 5.7. Carbonates and sulphates are the essential ions for possible precipitation or scaling and carbonate ions should also be considered here.”

The carbonate ions are essential, but they are also easier to remove in the plant pretreatment (i.e. by decarbonization using acid dosing or weak ion exchangers). As such, we decided to focus on more difficult scale-forming compound, calcium sulphate.

“Chlorides, for example, are highly soluble and seems to me meaningless in such a study, as they only make a small contribution to flow resistance and osmotic pressure. Why are chlorides also described here.”

The chlorides themselves don’t create insoluble salts in the tested conditions, but they highly affect the ionic strength, especially in waters which contain mainly chlorides; because of that, the solubility of calcium sulphate varies in brines of different salinity and the omission of chlorides would make the saturation calculations unreliable.

“The notion of a recovery Y, to which the retention in the test cell is referred to here, seems unusual for such tests. A factor that reflects the volumetric concentration of the feed solution introduced into the test cell should be more appropriate here.”

We’ve added volume reduction factor along with recovery to the Table A1 and included Equation 41, linking VRF and Y.

“Furthermore, the specific permeate flows to be observed in these measurements of the retention should be indicated.”

We’ve added the observed permeate flows in the Appendix A.

“With a membrane area of 14.6 cm2 and a pressure of 40 bar I estimate these permeate flows in the range of 1 mL/min. The starting volume in the test cell is also not given. If 100 mL were to be filled, such a concentration experiment, in which 90 % of the feed solution is obtained as permeate, would take about 2 h, whereby the concentrations in the retentate change over time. In a technical plant for water treatment, the operation is normally continuous and the concentrations and flows are locally constant. The experimental data are taken here from discontinuous unsteady experiments. It has to be explained why these data are suitable for statements on continuous separation processes and here for the assessment of the maximum possible recovery in operating plants. “

It is true that this not perfect and accurate simulation of the conditions happening inside an industrial spiral-wound module. The purpose of the proposed method was to estimate the borderline conditions for the high-recovery nanofiltration of the waters having high scaling risk. For this purpose no one is going to risk operating industrial-scale modules at 80-90% recovery and the bench-scale experiments are always unsteady-state in practice (e.g. to reach 80% using standard SEPA module, recirculation would be necessary, which would defy the whole purpose of trying to work in steady-state single-pass mode). Based on presented estimations we can design and build a pilot-scale nanofiltration and set the framework for more accurate testing: which membrane to use, what recovery limit we should not cross in large-scale tests, how to position scaling detectors (e.g. ultrasonic ones) along the module for the detection of scaling onset. We have included the issues you raised in the revised manuscript.

“Line 85:  heady load ?”

Changed to “large amount”

“Line 22: twice NF270 !”

Should be NFDL instead of second NF270, fixed.

“Line 67: NFDL-5 is originally a Desal-Membrane, than GE-Osmonics and now a Suez membrane product”

Replaced “GE Osmonics” with “Suez”.

“Line 158: Why it's a Stirred Cell membrane module. Such a stirred cell contains a flat membrane and a membrane module is a technical concept with a specific configuration for the application.”

We have used batch-mode stirred cell module because it is very easy to reach high recovery (80-90%) while simultaneously avoiding concentration by high speed mixing (1200 rpm) and maintain constant temperature with a cooling jacket.

“Line 203 to 206: Here the laboratory data are transferred to a continuous separation process with a 40” spiral module (as stated above). How is this module operated, what is the procedure here and why is this permissible or acceptable and applicable?”

We simulate the spiral-wound module operated in the single-pass mode, without the recirculation. We assumed that the recovery in this module is controlled by the feed flow rate. We assumed the rejection coefficients obtained in the batch-mode studies would be applicable in the modelling of single-pass operation, as long as we assure negligible concentration polarization conditions; the feed conditions and resulting rejection coefficients/fluxes changing in time would simulate the feed conditions changing along the membrane in the module.

“Line 279: I don't think that a redesign of the module is the task here.

Replaced with “a method of predicting the operational limits…”

“Line 282: …it is possible to determine possibly…seems to me a speculation and not a conclusion!”

Replaced with “it is possible to predict”

“Line 287: Explain the overall recovery newly mentioned here. Why this extrapolation is possible. It seems more like speculation here.”

What we mean is that if nanofiltration is used as a pretreatment for other methods (i.e. reverse osmosis or distillation), the low permeate recovery in the NF part limits the recovery of the whole technology train (overall recovery). We rephrased the sentence in question: “A possibility of working at 75% permeate recovery level should improve the performance of the integrated salt production systems using NF as pretreatment step, since they are limited in terms of overall recovery by the pretreatment (NF) recovery.”

“Line 291: “…possible precise…”  Is it or not and what means then precise here.”

We meant this model is as precise as possible given all the uncertainties of the experiment. Removed as not that important for the overall conclusions.

Reviewer 2 Report

membranes‐917196

Scaling risk assessment in nanofiltration of mine waters

The work is describing the estimation of the scaling of gypsum onto NF when treating two types of brines from coal mines. The prediction was based additionally from mas transfer parameters and modules design on the dependence of ions rejection as a function of water recover. The predictions developed are directed to propose a method for estimating the maximum achievable recovery before the occurrence of the calcium sulphate dehydrate scaling in a single‐pass spiral wound module. Three different types of NF membranes were evaluated. The experimental methodology and the data treatment methods used are sound.

The work is inside of the journal although not too much membrane science is covered and the major contribution is related to more practical and operation aspects. The novelty of the work is limited as prediction of scaling of minerals as gypsum, is done by most of the projection tools of most of the membrane providers (Rosa from Dow, Winflos from GE‐Suez, or the Hydranautics or Lanxess). Then, it is needed a discussion from the authors on describing the innovation of the manuscript. What it is the difference with the scaling prediction tolls described.

Additionally, there are a series of issues that need attention:

‐ A main criticism is the lack of the statistical quality criteria required by the journal: any of the data in figures, tables or along the text are provided without error bars and without confidence level.

‐ Rejection data were modelled as a function of the water recovery, but the dependence of the transmembrane pressure difference is very important. Then, the model will be limited for the given pressure selected.

‐ The predictions identified scenarios of scaling but not effort was allocated to analysed the membrane surface at the end of the filtration trials. Checking this possibility, it is a lost opportunity.

‐ The different performance on membrane rejection as a function of the membrane properties was not analysed. Also the chemical properties of the active layer as the chemical nature of the PA, and the isoelectric point.

‐ Which mechanisms explain the different performance of ions rejections in both types of brines, the influence of the solution composition has not been discussed.

Other Minor comments:

‐ Keywords (nanofiltration; scaling; coal mine waters; modelling): please do not include those already present in the title.

‐ Figure captions and table headings need attention, as they should be understood totally without reading the test. As an example brine A or brine B, should be accompanied by the composition of those brines.

‐ In the membrane description table, the information provided is not relevant for the paper and it is lacking the relevant one.

Author Response

Thank you for the thorough review. We hope we fixed all the problems you pointed out. The changes are highlighted in the manuscript, please find below the detailed response to your questions.

“The novelty of the work is limited as prediction of scaling of minerals as gypsum, is done by most of the projection tools of most of the membrane providers (Rosa from Dow, Winflos from GE Suez, or the Hydranautics or Lanxess). Then, it is needed a discussion from the authors on describing the innovation of the manuscript. What it is the difference with the scaling prediction tolls described.”

The software provided by the manufacturers typically predict final parameters of the retentate/permeate, however modelling the nanofiltration in the manner presented in the manuscript can give additional information, such as concentration profile along the membrane, the place inside the module where the scaling risk increase.

“A main criticism is the lack of the statistical quality criteria required by the journal: any of the data in figures, tables or along the text are provided without error bars and without confidence level”

We have calculated the error in rejection assuming each of the ionic chromatography concentration measurements had 3% instrumental error. The gypsum saturation data are purely computational and relies partly on rejection coefficients calculated based on our experimental data and partly on empirical parameters given in the literature without specifying the error bars (e.g. for the Bromley equation).

“Rejection data were modelled as a function of the water recovery, but the dependence of the transmembrane pressure difference is very important. Then, the model will be limited for the given pressure selected.”

We agree and we added explicit statement that the results are valid for 40 bar of operating pressure.

“The predictions identified scenarios of scaling but not effort was allocated to analysed the membrane surface at the end of the filtration trials. Checking this possibility, it is a lost opportunity.”

The membrane autopsy and the effect of filtration on the membrane surface was not a scope of this study.

“The different performance on membrane rejection as a function of the membrane properties was not analysed. Also the chemical properties of the active layer as the chemical nature of the PA, and the isoelectric point. Which mechanisms explain the different performance of ions rejections in both types of brines, the influence of the solution composition has not been discussed.”

We’ve added the discussion on possible reason behind differences in rejection coefficients: “The rejection coefficients for brackish water “A” are considerably higher - for all ions considered - than for brine “B”. This may be caused by the different composition of each feed water. The water “A” contains less chloride, sodium, calcium, and magnesium ions, which causes smaller diffusion driving force across the membrane, resulting in lower flux of these ions across the membrane; simultaneously, the water flux across the membrane is higher in the case of less saline water “A”, causing overall lower rejection coefficients of calcium and magnesium. This effect doesn’t happen the case of the sulphate, as both waters contain similar amounts of this ion. The rejection of sulfate was significantly higher than the rejection of calcium and magnesium, which was caused by the negative surface charge of the membrane at the experimental conditions (pH ≥ 5.7).

The highest chlorides rejection was observed when the NFDL membrane was used. Moreover, in most cases NDFL membrane type demonstrates the highest rejection of both univalent cations and bivalent cations, while NFX membrane shows generally the lowest rejection among all 3 types studied. ”

“Keywords (nanofiltration; scaling; coal mine waters; modelling): please do not include those already present in the title.”

We’ve chosen different keywords.

“Figure captions and table headings need attention, as they should be understood totally without reading the test. As an example brine A or brine B, should be accompanied by the composition of those brines.”

Fixed figure captions and table headings.

“In the membrane description table, the information provided is not relevant for the paper and it is lacking the relevant one.”

We’ve added the isoelectric point and MWCO to the description of table properties.

Reviewer 3 Report

Although the writing and the quality of the experiments seem to be of decent quality I have one major concern and several remarks. In my opinion these need to be addressed before the manuscript could be considered for publication.

My main concern is whether the use of dead-end filtration in the first part of the manuscript is appropriate here. You use feed solutions with high salt concentrations which are thus extremely prone to concentration polarization. Dead-end filtration is additionally much more prone to concentration polarization than crossflow. You stir to somewhat alleviate this but you simply do not know to which extent concentration polarization plays in your results and even more so if and to what extent it varies during your recovery experiment. These rejection numbers can in my opinion therefore not simply be used in your modeling. In the modelling you then add a term for concentration polarization so then you have taken this into account twice? One time inherent from your experiment and one time from the modeling? I don’t think this is correct. You should think of a more correct way to get these rejection as function of recovery numbers. Moreover, I think such experiments belong in the supporting information as they are not novel and only provide the input parameters for the model.

A major practical issue I have with the manuscript is the quality of the figures. They have a very unprofessional look to them. (i) The frame surrounding the figures is unnecessary. (ii) In some cases there are both horizontal and vertical guidelines whereas in others only horizontal. (iii) In some cases the axis titles are words, in others just symbols. I would suggest the authors to find a standardized look to their figures and to think about what is necessary for good readability of the figure; e.g. the vertical guide lines are not necessary in my opinion as this separation between points is clear enough. Perhaps a bit of color could also help the viewing of the figures. Finally I would darken the tint of grey that is used. I know pure black is sometimes considered not good for the eyes but this type of lighter grey almost looks like it was photocopied.

I would suggest the authors check whether all references are appropriate. E.g. in line 30, they refer to reference 11 as NF being used in the recovery of proteins and sugars. This paper does not do that. They separate proteins from sugars (mind that separate does not equal recover) and this using UF membranes! Non-accurate referencing is a prerequisite for any paper.

It is not clear whether mine water A and B were artificial or collected somewhere. Please specify and provide more information.

Minor remarks

Line 27-28: I don’t think the first sentence of the introduction requires 12 references. A few research articles and/or reviews are sufficient.

Line 28-29: NF is generally quoted to be very selective to all ions except for monovalent ones. I would suggest to write it in this general way because this way you almost suggest NF is particularly good at sulfates/Ca/Mg whereas.

Line 35-37: I don’t see why you specifically write calcium sulphate as one of the possible hydrates and the others not?

Line 44: I agree with what you write, don’t get me wrong, but the use of ‘therefore’ is not good here. This suggests you need accurate prediction because of the anti-scaling strategies that you mention in the previous sentences.

Line 176-199: Personally I find this discussion not adding much more information than what is already in the figure and this is information that is already well documented in literature. I would shorten this discussion to summarize the key ‘take-home-information’. Am I correct that the only reason for the experiments in figure 2-5 is to get the rejection at various recovery for the different membranes and ions? If so this is not the main point of the article and should be moved to the supporting information.

Author Response

Thank you for the thorough review. We tried to address the issues best we can and we hope the quality of the manuscript is now better. We highlighted all the changes in the revised version of the manuscript. Please find below the detailed response to the issues you pointed out.

“My main concern is whether the use of dead-end filtration in the first part of the manuscript is appropriate here. You use feed solutions with high salt concentrations which are thus extremely prone to concentration polarization. Dead-end filtration is additionally much more prone to concentration polarization than crossflow. You stir to somewhat alleviate this but you simply do not know to which extent concentration polarization plays in your results and even more so if and to what extent it varies during your recovery experiment. These rejection numbers can in my opinion therefore not simply be used in your modeling. In the modelling you then add a term for concentration polarization so then you have taken this into account twice? One time inherent from your experiment and one time from the modeling? I don’t think this is correct. You should think of a more correct way to get these rejection as function of recovery numbers. Moreover, I think such experiments belong in the supporting information as they are not novel and only provide the input parameters for the model.”

We’ve moved the results of the experiments to the Appendix section. As for the concentration polarization problems, we think the results obtained in our dead-end filtration could be used for modelling the cross-flow module as long as we ensure concentration polarization during studies was negligible, so we observe fluxes through the membrane alone and don’t count the CP twice as you pointed out. To do that, we have used very high speed mixer (1200 rpms), which together with the dimensions of the system, resulted in turbulent flow, not prone to concentration polarization (Re = 63 200, 700 < Sc < 2100, 1500 < Sh < 2200). We think these kind of flow conditions assure the rejection coefficients we measured were not affected by the CP. We added these considerations to the manuscript, as well as literature reference to the paper discussing the effect of hydrodynamic conditions on the rejection coefficients: „The idea is to use the rejection coefficients observed in the dead-end filtration in the modelling of the cross-flow filtration. The necessary condition is to create hydrodynamic conditions which assure the concentration polarization is negligible and does not affect the conditions during the batch studies. For instance, in other study using the dead-end filtration it was observed that in the stirred cell the concentration polarization became negligible at rpm > 100 and Re > 12 000 [18]. To assure lack of concentration polarization, we have used very high speed mixer (1200 rpms), which together with the dimensions of the system, resulted in turbulent flow and good mixing near the membrane surface (Re = 63 200, 700 < Sc < 2100, 1500 < Sh < 2200).”

“A major practical issue I have with the manuscript is the quality of the figures. They have a very unprofessional look to them. (i) The frame surrounding the figures is unnecessary. (ii) In some cases there are both horizontal and vertical guidelines whereas in others only horizontal. (iii) In some cases the axis titles are words, in others just symbols. I would suggest the authors to find a standardized look to their figures and to think about what is necessary for good readability of the figure; e.g. the vertical guide lines are not necessary in my opinion as this separation between points is clear enough. Perhaps a bit of color could also help the viewing of the figures. Finally I would darken the tint of grey that is used. I know pure black is sometimes considered not good for the eyes but this type of lighter grey almost looks like it was photocopied.”

We worked on the figures and hopefully they are of better quality now.

“I would suggest the authors check whether all references are appropriate. E.g. in line 30, they refer to reference 11 as NF being used in the recovery of proteins and sugars. This paper does not do that. They separate proteins from sugars (mind that separate does not equal recover) and this using UF membranes! Non-accurate referencing is a prerequisite for any paper.”

We fixed errors in the references.

“It is not clear whether mine water A and B were artificial or collected somewhere. Please specify and provide more information.”

Both waters were collected at the mines, but since we don’t have permission from the companies running these operations to state their names explicitely. A is from abandoned zinc mine “Orzeł Biały” in Bytom, Poland (after the zinc has been effectively removed by Ca(OH)2 + sedimentation in a pond, so its presence was no longer an issue, but salinity was exceeding tap water limits), B is from active coal mine “Budryk” in Ornontowice, Poland. We specificied in the text that these were real samples.

Minor remarks

“Line 27-28: I don’t think the first sentence of the introduction requires 12 references. A few research articles and/or reviews are sufficient.”

Removed unnecessary references.

“Line 28-29: NF is generally quoted to be very selective to all ions except for monovalent ones. I would suggest to write it in this general way because this way you almost suggest NF is particularly good at sulfates/Ca/Mg whereas.”

Replaced with: “Nanofiltration, due to its high rejection to all ions except the monovalent ones, is widely used…“

“Line 35-37: I don’t see why you specifically write calcium sulphate as one of the possible hydrates and the others not?”

Added other forms of calcium sulphate.

“Line 44: I agree with what you write, don’t get me wrong, but the use of ‘therefore’ is not good here. This suggests you need accurate prediction because of the anti-scaling strategies that you mention in the previous sentences.”

Removed the misleading wording.

“Line 176-199: Personally I find this discussion not adding much more information than what is already in the figure and this is information that is already well documented in literature. I would shorten this discussion to summarize the key ‘take-home-information’. Am I correct that the only reason for the experiments in figure 2-5 is to get the rejection at various recovery for the different membranes and ions? If so this is not the main point of the article and should be moved to the supporting information.”

We’ve moved the graphs to the appendix and made discussion of the results in section 3.1 more concise.

Round 2

Reviewer 2 Report

In the

The new version describes a big effort to cover comments requested on my review, and also the comments and suggestion of the two other reviewers. It is of mention that even am extended review of requested topics has been performed. In most of the requests consistent answers were provided and were accompanied even high new information.  All requested additional details were also introduced in the text and in some cases sections has been re-structured. Also some of the information has been sent as supplementary to reach the journal recommendations o manuscript length.

The quality and the clarity of the paper message have been really improved and it is collecting, to my opinion, a valuable text on prediction of scaling on NF processes.

Author Response

Thank you for reviewing the revised version of the manuscript.

(There was no follow-up questions from this reviewer)

Reviewer 3 Report

I think appropriate changes to the manuscript were made and I think therefore it can be considered for publication.

I have few remarks remaining regarding the concentration polarization that I think should be changed. (1) The authors mention the work from Prof. Schaefer regarding CP and Re-number/rotation speed. It is important to clarify in the text that such numbers are really depending on the type of setup that is used. 100 rpm in one setup with a certain dimension and type/shape/dimension of stirring bar is not the same as in a different setup. (2) I think it would be great if you put the calculations for the mentioned Re Sc and Sh numbers in the supporting information so readers can see how you end up with these numbers and where the ranges come from. (3) Given that a flow profile over the membrane surface is never homogeneous in dead-end filtration, I think it is important to always say 'minimized concentration polarization' not 'lack of CP', in dead-end there is never no CP.

Author Response

Thank you for reviewing the changes we made to the initial version of the manuscript. We have addressed your follow-up questions below:

"(1) The authors mention the work from Prof. Schaefer regarding CP and Re-number/rotation speed. It is important to clarify in the text that such numbers are really depending on the type of setup that is used. 100 rpm in one setup with a certain dimension and type/shape/dimension of stirring bar is not the same as in a different setup."

We have added in the text the explanation that at the high Re the design of the mixer makes a big difference and value of rpms at which the CP is minimized also depends on type/shape of mixer.

"(2) I think it would be great if you put the calculations for the mentioned Re Sc and Sh numbers in the supporting information so readers can see how you end up with these numbers and where the ranges come from."

We have added the Re, Sc, Sh calculations to the Appendix.

"(3) Given that a flow profile over the membrane surface is never homogeneous in dead-end filtration, I think it is important to always say 'minimized concentration polarization' not 'lack of CP', in dead-end there is never no CP."

We have changed "no CP" to "minimized CP".